# The lawful imprecision of human surface tilt estimation in natural scenes

Seha Kim*, Johannes Burge*

Department of Psychology, University of Pennsylvania, Philadelphia, United States

**Abstract** Estimating local surface orientation (slant and tilt) is fundamental to recovering the three-dimensional structure of the environment. It is unknown how well humans perform this task in natural scenes. Here, with a database of natural stereo-images having groundtruth surface orientation at each pixel, we find dramatic differences in human tilt estimation with natural and artificial stimuli. Estimates are precise and unbiased with artificial stimuli and imprecise and strongly biased with natural stimuli. An image-computable Bayes optimal model grounded in natural scene statistics predicts human bias, precision, and trial-by-trial errors without fitting parameters to the human data. The similarities between human and model performance suggest that the complex human performance patterns with natural stimuli are lawful, and that human visual systems have internalized local image and scene statistics to optimally infer the three-dimensional structure of the environment. These results generalize our understanding of vision from the lab to the real world.

DOI: https://doi.org/10.7554/eLife.31448.001

## Introduction

Understanding how vision works in natural conditions is a primary goal of vision research. One measure of success is the degree to which performance in a fundamental visual task can be predicted directly from image data. Estimating the 3D structure of the environment from 2D retinal images is just such a task. However, relatively little is known about how the human visual system estimates 3D surface orientation from images of natural scenes.

3D surface orientation is typically parameterized by slant and tilt. Slant is the amount by which a surface is rotated away from an observer; tilt is the direction in which the surface is rotated (*Figure 1A*). Compared to slant, tilt has received little attention, even though both are critically important for successful interaction with the 3D environment. For example, even if slant has been accurately estimated, humans must estimate tilt to determine where they can walk. Surface with tilts of 90°, like the ground plane, can sometimes be walked on. Surfaces with tilts of 0° or 180°, like the sides of tree trunks, can never be walked on.

Numerous psychophysical, computational, and neurophysiological studies have probed the human ability to estimate surface slant, surface tilt, and 3D shape. Systematic performance has been observed, and models have been developed that nicely describe performance. Most previous studies have used stimuli having planar (*Stevens, 1983*; *Knill, 1998a, 1998b*; *Hillis et al., 2004*; *Burge et al., 2010a*; *Rosenholtz and Malik, 1997*; *Rosenberg et al., 2013*; *Murphy et al., 2013*; *Velisavljević and Elder, 2006*; *Saunders and Knill, 2001*; *Welchman et al., 2005*; *Sanada et al., 2012*; *Tsutsui et al., 2001*) or smoothly curved (*Todd et al., 1996*; *Fleming et al., 2011*; *Todd, 2004*; *Marlow et al., 2015*; *Li and Zaidi, 2000, 2004*; *Norman et al., 2006*) surface shapes and regular (*Knill, 1998a, 1998b*; *Hillis et al., 2004*; *Watt et al., 2005*; *Rosenholtz and Malik, 1997*; *Rosenberg et al., 2013*; *Murphy et al., 2013*; *Velisavljević and Elder, 2006*; *Li and Zaidi, 2000, 2004*; *Welchman et al., 2005*) or random-patterned (*Burge et al., 2010a*; *Fleming et al., 2011*) surface markings. These stimuli are not representative of the variety of surface shapes and

**\*For correspondence:**
sehakim@upenn.edu (SK);
jburge@sas.upenn.edu (JB)

**Competing interests:** The authors declare that no competing interests exist.

**eLife digest** The ability to assess how tilted a surface is, or its 'surface orientation', is critical for interacting productively with the environment. For example, it helps organisms to determine whether a particular surface is better suited for walking or climbing. Humans and other animals estimate 3-dimensional (3D) surface orientations from 2-dimensional (2D) images on their retinas. But exactly how they calculate the tilt of a surface from the retinal images is not well understood.

Scientists have studied how humans estimate surface orientation by showing them smooth (often planar) surfaces with artificial markings. These studies suggested that humans very accurately estimate the direction in which a surface is tilted. But whether humans are as good at estimating surface tilt in the real world, where scenes are more complex than those tested in experiments, is unknown.

Now, Kim and Burge show that human tilt estimation in natural scenes is often inaccurate and imprecise. To better understand humans' successes and failures in estimating tilt, Kim and Burge developed an optimal computational model, grounded in natural scene statistics, that estimates tilt from natural images. Kim and Burge found that the model accurately predicted how humans estimate tilt in natural scenes. This suggests that the imprecise human estimates are not the result of a poorly designed visual system. Rather, humans, like the computational model, make the best possible use of the information images provide to perform an estimation task that is very difficult in natural scenes.

The study takes an important step towards generalizing our understanding of human perception from the lab to the real world.

DOI: https://doi.org/10.7554/eLife.31448.002

markings encountered in natural viewing. Surfaces in natural scenes often have complex surface geometries and are marked by complicated surface textures. Thus, performance with simple artificial scenes may not be representative of performance in natural scenes. Also, models developed with artificial scenes often generalize poorly (or cannot even be applied) to natural scenes. These issues concern not just studies of 3D surface orientation perception but vision and visual neuroscience at large.

Few studies have examined the human ability to estimate 3D surface orientation using natural photographic images, the stimuli that our visual systems evolved to process. None, to our knowledge, have done so with high-resolution groundtruth surface orientation information. There are good reasons for this gap in the literature. Natural images are complex and difficult to characterize mathematically, and groundtruth data about natural scenes are notoriously difficult to collect. Research with natural stimuli has often been criticized (justifiably) on the grounds that natural stimuli are too complicated or too poorly controlled to allow strong conclusions to be drawn from the results. The challenge, then, is to develop experimental methods and computational models that can be used with natural stimuli without sacrificing rigor and interpretability.

Here, we report an extensive examination of human 3D tilt estimation from local image information with natural stimuli. We sampled thousands of natural image patches from a recently collected stereo-image database of natural scenes with precisely co-registered distance data (*Figure 1B*) (*Burge et al., 2016*). Groundtruth surface orientation was computed directly from the distance data (see Materials and methods). Human observers binocularly viewed the natural patches and estimated the tilt at the center of each patch. The same human observers also viewed artificially-textured planar stimuli matched to the groundtruth tilt, slant, distance, and luminance contrast of the natural stimuli. First, we compared human performance with natural and matched artificial stimuli. Then, we compared human performance to the predictions of an image-computable normative model, a Bayes' optimal observer, that makes the best possible use of the available image information for the task. This experimental design enables direct, meaningful comparison of human performance across stimulus types, allowing the isolation of important stimulus differences and the interpretation of human response patterns with respect to principled predictions provided by the model.

A rich set of results emerges. First, tilt estimation in natural scenes is hard; compared to performance with artificial stimuli, performance with natural stimuli is poor. Second, with natural stimuli,

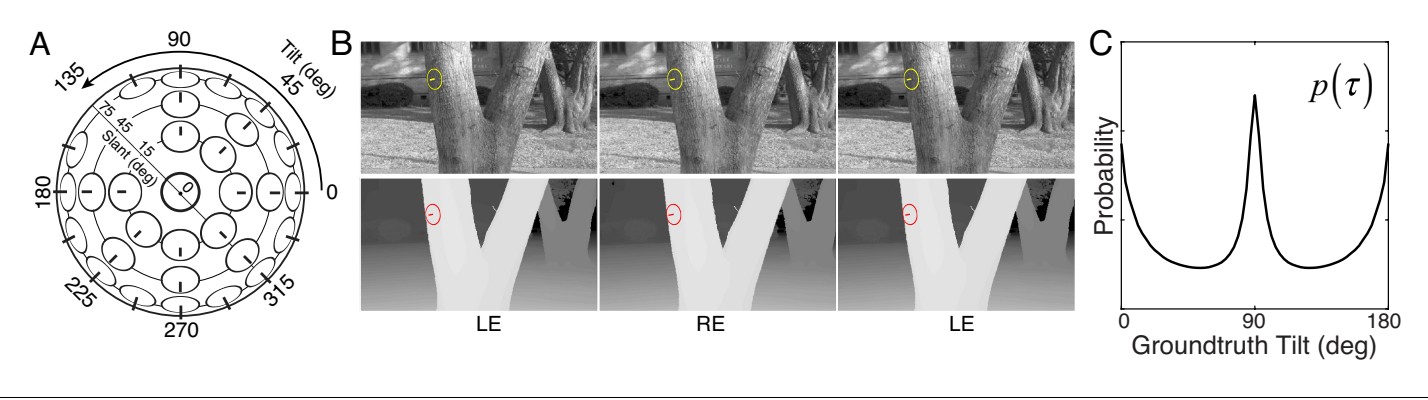

**Figure 1.** Tilt and slant, natural scene database, and tilt prior. (A) Tilt is the direction of slant. Slant is the amount of rotation out of the reference (e.g., frontoparallel) plane. (B) Example stereo-image pair (top) and corresponding stereo-range data (bottom). The gauge figure indicates local surface orientation. To see the scene in stereo 3D, free-fuse the left-eye (LE) and right-eye (RE) images. (C) Prior distribution of unsigned tilt in natural scenes, computed from 600 million groundtruth tilt samples in the natural scene database (see Materials and methods). Cardinal surface tilts associated with the ground plane (90°) and tree trunks (0° and 180°) occur far more frequently than oblique tilts in natural scenes. Unsigned tilt, $\tau = [0°, 180°]$, indicates 3D surface orientation up to a sign ambiguity (i.e., tilt modulo 180°).

DOI: https://doi.org/10.7554/eLife.31448.003

human tilt estimates cluster at the cardinal tilts (0°, 90°, 180° and 270°), echoing the prior distribution of tilts in natural scenes (*Figure 1C*) (*Burge et al., 2016*; *Yang and Purves, 2003a*; *Yang and Purves, 2003b*; *Adams et al., 2016*). Third, human estimates tend to be more biased and variable when the groundtruth tilts are oblique (e.g., 45°). Fourth, at each groundtruth tilt, the distributions of human and model errors tend to be very similar, even though the error distributions themselves are highly irregular. Fifth, human and model observer trial-by-trial errors are correlated, suggesting that similar (or strongly correlated) stimulus properties drive both human and ideal performance. Together, these results represent an important step towards the goal of being able to predict human percepts of 3D structure directly from photographic images in a fundamental natural task.

## Results

Human observers binocularly viewed thousands of randomly sampled patches of natural scenes; they viewed an equal number of stimuli at each of 24 tilt bins between 0° and 360°. The stimuli were presented on a large (2.0 × 1.2 m) stereo front-projection system positioned 3 m from the observer. This relatively long viewing distance minimizes focus cues to flatness. Except for focus cues, the display system recreates the retinal images that would have been formed by the original scene. Each scene was viewed binocularly through a small virtual aperture (1° or 3° of visual angle) positioned 5 arcmin of disparity in front of the sampled point in the scene (*Figure 2A*); the viewing situation is akin to looking at the world through a straw (*McDermott, 2004*). Patches were displayed at the random image locations from which they were sampled. Observers reported, using a mouse-controlled probe, the estimated surface tilt at the center of each patch (*Figure 2B*). We pooled data across human observers and aperture sizes and converted the tilt estimates to unsigned tilt for analysis (signed tilt modulo 180°) because the estimation of unsigned tilt was similar for all observers and aperture sizes (*Figure 2—figure supplement 1*, *Figure 2—figure supplement 2*). The same observers also estimated surface tilt with an extensive set of artificial planar stimuli that were matched to the tilts, slants, distances, and luminance contrasts of the natural stimuli presented in the experiment. (Each planar artificial stimulus had one of three texture types: 1/f noise, 3.5 cpd plaid, and 5.25 cpd plaid; *Figure 2—figure supplement 3*.) Thus, any observed performance differences between natural and artificial stimuli cannot be attributed to these dimensions.

Natural and artificial stimuli elicited strikingly different patterns of performance (*Figure 2C*). Although many stimuli of both types elicit tilt estimates $\hat{\tau}$ that approximately match the groundtruth tilt (data points on the unity line), a substantial number of natural stimuli elicit estimates that cluster at the cardinal tilts (data points at $\hat{\tau} = 0°, 90°, 180°, 270°$). No such clustering occurs with artificial

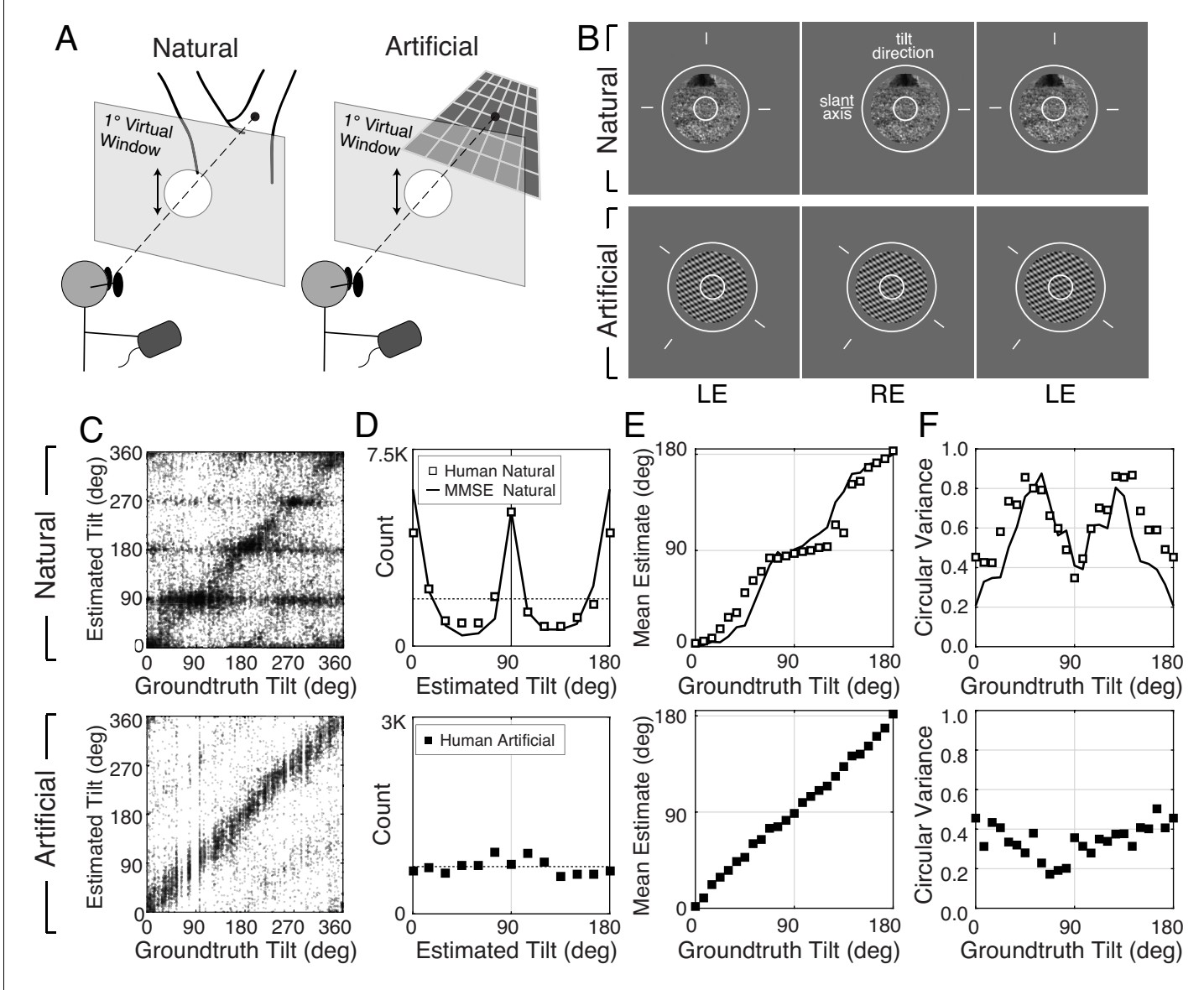

**Figure 2.** Experimental stimuli and human tilt responses. (A) The virtual viewing situation. (B) Example natural stimulus (ground plane) and artificial stimulus (3.5cpd plaid). See *Figure 2—figure supplement 3* for all three types. The task was to report the tilt at the center of the small (1° diameter) circle. Aperture sizes were either 3° (shown) or 1° (not shown) of visual angle. Observers set the orientation of the probe (circle and line segments) to indicate estimated tilt. Free-fuse to see in stereo 3D. (C) Raw responses for every trial in the experiment. (D) Histogram of raw responses (unsigned estimates). The dashed horizontal line shows the uniform distribution of groundtruth tilts presented in the experiment. (Histograms of signed tilt estimates are shown in *Figure 2—figure supplement 4*.) (E) Estimate means and (F) estimate variances as a function of groundtruth tilt. Human tilt estimates are more biased and variable with natural stimuli (top) than with artificial stimuli (bottom). Data are combined across all three artificial texture types; see *Figure 2—figure supplement 3* for performance with each individual texture type. With artificial stimuli, human estimates are unbiased and estimate variance is low. Model observer predictions (minimum mean squared error [MMSE] estimates; black curves) parallel human performance with natural stimuli.

DOI: https://doi.org/10.7554/eLife.31448.004

The following figure supplements are available for figure 2:

**Figure supplement 1.** Tilt estimation errors with small vs. large apertures for natural stimuli.

DOI: https://doi.org/10.7554/eLife.31448.005

**Figure supplement 2.** Tilt estimation performance for individual human observers.

DOI: https://doi.org/10.7554/eLife.31448.006

**Figure supplement 3.** Summary statistics for the three different artificial stimulus types: 1/f noise, 3.50 cpd plaid, 5.25 cpd plaid (top, middle, and bottom rows, respectively).

*Figure 2 continued on next page*

*Figure 2 continued*

DOI: https://doi.org/10.7554/eLife.31448.007

**Figure supplement 4.** Histogram of raw responses (estimates) from human observers in the signed tilt domain $\tau = [0°, 360°)$.

DOI: https://doi.org/10.7554/eLife.31448.008

**Figure supplement 5.** Tilt estimation performance with full-field (36° x 21°) viewing of natural stimuli.

DOI: https://doi.org/10.7554/eLife.31448.009

stimuli. The histogram of the human tilt estimates explicitly shows the clustering, or lack thereof (*Figure 2D*). With natural stimuli, the distribution of unsigned estimates $p(\hat{\tau})$ peaks at 0° and 90° and has a similar shape to the prior distribution of groundtruth tilts in the natural scene database (*Figure 1C*; also see *Figure 2—figure supplement 4*). If the database is representative of natural scenes, then one might expect the human visual system to use the natural statistics of tilt as a tilt prior in the perceptual processes that convert stimulus measurements into estimates. Standard Bayesian estimation theory predicts that the prior will influence estimates more when measurements are unreliable and will influence estimates less when measurements are reliable (*Knill and Richards, 1996*).

We summarized 3D tilt estimation performance by computing the mean and variance of the tilt estimates $\hat{\tau}$ as a function of groundtruth tilt (*Figure 2E,F*). (The mean and variance were computed using circular statistics because tilt is an angular variable; see Materials and methods.) These summary statistics change systematically with groundtruth tilt, exhibiting patterns reminiscent of the 2D oblique effect (*Appelle, 1972*; *Furmanski and Engel, 2000*; *Girshick et al., 2011*). With natural stimuli, estimates are maximally biased at oblique tilts and unbiased at cardinal tilts; estimate variance is highest at oblique tilts (~60° and ~120°) and lowest at cardinal tilts. With artificial stimuli, estimates are essentially unbiased and are less variable across tilt. The unbiased responses to artificial stimuli imply that the biased responses to natural stimuli accurately reflect biased perceptual estimates, under the assumption that the function that maps perceptual estimates to probe responses is stable across stimulus types (see Materials and methods). (See *Figure 2—figure supplement 3* for performance with each individual artificial texture type.) The summary statistics reveal clear differences between the stimulus types. However, there is more to the data than the summary statistics can reveal. Thus, we analyzed the raw data more closely.

The probabilistic relationship between groundtruth tilt $\tau$ and human tilt estimates $\hat{\tau}$ is shown in *Figures 3* and *4*. Each subplot in *Figure 3A* shows the distribution of estimation errors $p(\hat{\tau} - \tau | \tau)$ for a different groundtruth tilt. With artificial stimuli, estimation errors $e = \hat{\tau} - \tau$ are unimodally distributed and peaked at zero (black symbols). With natural stimuli, estimation errors are more irregularly distributed, and the peak locations change systematically with groundtruth tilt (white points). With cardinal groundtruth tilts (e.g., $\tau = 0°$ or $\tau = 90°$), the error distributions peak at zero and large errors are rare. With oblique groundtruth tilts (e.g., $\tau = 60°$ or $\tau = 120°$), the error distributions tend to be bi-modal with two prominent peaks at non-zero errors. For example, when groundtruth tilt $\tau = 60°$, the most common errors were $-60°$ and 30°. These errors occurred because observers incorrectly estimated the tilt to be 0° or 90°, respectively, when the correct answer was 60°. Thus, at this groundtruth tilt, the human observers frequently (and incorrectly) estimated cardinal tilts instead of the correct oblique tilt.

Tilt estimates from natural stimuli are less accurate at oblique than at cardinal groundtruth tilts. Does this fact imply that oblique tilt estimates (e.g., $\hat{\tau} = 60°$) provide less accurate information about groundtruth tilt than cardinal tilt estimates (e.g., $\hat{\tau} = 90°$)? No. Each panel in *Figure 4A* shows the distribution of groundtruth tilts $p(\tau | \hat{\tau})$ for each estimated tilt. The most probable groundtruth tilt equals the estimated tilt, and the variance of each distribution is approximately constant, regardless of whether the estimated tilt is cardinal or oblique. Furthermore, the estimates from natural and artificial stimuli provide nearly equivalent information about groundtruth (see also *Figure 4—figure supplement 1*). Thus, even though tilt estimation performance is far poorer at oblique than at cardinal tilts and is far poorer with natural than with artificial stimuli, all tilt estimates regardless of the value are similarly good predictors of groundtruth tilt.

How can it be that low-accuracy estimates from natural stimuli predict groundtruth nearly as well as high-accuracy estimates from artificial stimuli? Some regions of natural scenes yield high-reliability measurements that make tilt estimation easy; other regions of natural scenes yield low-reliability

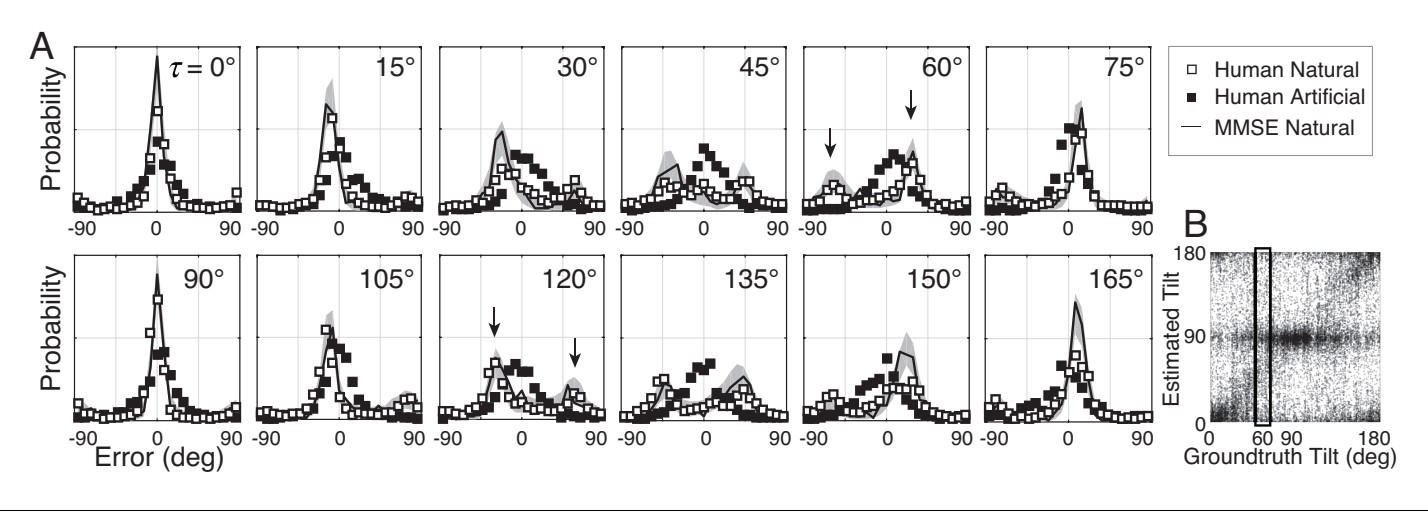

**Figure 3.** Distribution of tilt estimation errors for different groundtruth tilts. (A) Conditional error distributions $p(\hat{\tau} - \tau|\tau)$ are obtained by binning estimates for each groundtruth tilt (vertical bins in *Figure 3B*) and subtracting the groundtruth tilt. With artificial stimuli, the error distributions are centered on 0° (black symbols). With natural stimuli, the error distributions change systematically with groundtruth tilt (white symbols). For cardinal groundtruth tilts (0° and 90°), the most common error is zero. For oblique tilts, the error distributions peak at values other than zero (e.g., arrows in $\tau = 60°$ and $\tau = 120°$ subplots). The irregular error distributions are nicely predicted by the MMSE estimator (black curve); shaded regions show 95% confidence intervals on the MMSE estimates from 1000 Monte Carlo simulations of the experiment (see Materials and methods). The MMSE estimator predicts human performance even though zero free parameters were fit to the human responses. (B) Raw unsigned tilt estimates with natural stimuli (same data as *Figure 2C*, but shown in the unsigned tilt domain). The rectangular box shows estimates in the $\tau = 60°$ tilt bin.
DOI: https://doi.org/10.7554/eLife.31448.010

measurements that make tilt estimation hard. When measurements are reliable, the prior influences estimates less; when measurements are unreliable, the prior influences estimates more. Thus, cardinal tilt estimates can result either from reliable measurements of cardinal tilts or from unreliable measurements of oblique tilts. On the other hand, oblique tilt estimates can only result from reliable measurements of oblique tilts, because the measurements must be reliable enough to overcome the influence of the prior. All these factors combine to make each tilt estimate, regardless of its value, an equally reliable predictor of groundtruth tilt. The uniformly reliable information provided by the estimates about groundtruth (see *Figure 4A*) may simplify the computational processes that optimally pool local estimates into global estimates (see Discussion). The generality of this phenomenon across natural tasks remains to be determined. However, we speculate that it may have widespread importance for understanding perception in natural scenes, as well as in other circumstances where measurement reliability varies drastically across spatial location.

## Normative model

We asked whether the complicated pattern of human performance with natural stimuli is consistent with optimal information processing. To answer this question, we compared human performance to the performance of a normative model, a Bayes optimal observer that optimizes 3D tilt estimation in natural scenes given a squared error cost function (*Burge et al., 2016*). The model takes three local image cues $\mathbf{C}$ as input — luminance, texture, and disparity gradients — and returns the minimum mean squared error (MMSE) tilt estimate $\hat{\tau}_{MMSE}$ as output. (The MMSE estimate is the mean of the posterior probability distribution over groundtruth tilt given the measured image cues.)

To determine the optimal estimate for each possible triplet of cue values, we use the natural scene database. At each pixel in the database, the image cues are computed directly from the photographic images within a local area, and the groundtruth tilt is computed directly from the distance data (see Materials and methods; [*Burge et al., 2016*]). In other words, the model is 'image-computable': the model computes the image cues from image pixels and produces tilt estimates as outputs.

We approximate the posterior mean $E[\tau|\mathbf{C}] = \sum_\tau \tau p(\tau|\mathbf{C})$ by computing the sample mean of the groundtruth tilt conditional on each unique image cue triplet (*Figure 5A*). The result is a table, or

'estimate cube,' where each cell stores the optimal estimate $\hat{\tau}_{MMSE} = E[\tau|\mathbf{C}]$ for a particular combination of image cues (*Figure 5B*).

In the cue-combination literature, cues are commonly assumed to be statistically independent (*Ernst and Banks, 2002*). In natural scenes, it is not clear whether this assumption holds. Fortunately, the normative model used here is free of assumptions about statistical independence and the form of the joint probability distribution (see Discussion). Thus, our normative model provides a principled benchmark, grounded in natural scene statistics, against which to compare human performance.

We tested the model observer on the exact same set of natural stimuli used to test human observers (*Figure 5C*). The model observer predicts the overall pattern of raw human responses (see also *Figure 5—figure supplement 1*). More impressively, the model observer predicts the counts, means, and variances of the human tilt estimates (*Figure 2D–F*), the conditional error distributions (*Figure 3*), and the conditional groundtruth tilt distributions (*Figure 4*). The model explains a large proportion of the variance for all of these performance measures (*Figure 5D*). These results indicate that human visual system estimates tilt in accordance with optimal processes that minimize error in natural scenes. We conclude that the biased and imprecise human tilt estimates with natural stimuli are nevertheless lawful.

Two points are worth emphasizing. First, this model observer had no free parameters that were fit to the human data (*Burge et al., 2016*); instead, the model observer was designed to perform the task optimally given the three image cues. Second, the close agreement between human and model performance suggests that humans use the same cues (or cues that strongly correlate with those) used by the normative model (see Discussion).

## Trial-by-trial error

If human and model observers use the same cues in natural stimuli to estimate tilt, variation in the stimuli should cause similar variation in performance. Are human performance and model observer performance similar in individual trials? The same set of natural stimuli was presented to all observers. Thus, it is possible to make direct, trial-by-trial comparisons of the estimation errors that each observer made. If the properties of individual natural stimuli influence estimates similarly across observers, then observer errors across trials should be correlated. Accounting for trial-by-trial errors is one of the most stringent comparisons that can be made between model and human performance.

Natural stimuli do elicit similar trial-by-trial errors from human and model observers (*Figure 6A*). The model predicts trial-by-trial human errors far better than chance. We quantify the model-human similarity with the circular correlation coefficients of the trial-by-trial model and human estimates (*Figure 6B*). The correlation coefficients are significant. This result implies that the errors are systematically and reliably dependent on the properties of natural stimuli and that these properties affect human and model observers similarly.

However, because both human and model observers produced biased estimates with natural stimuli (*Figure 2E*, *Figure 2—figure supplement 2*), it is possible that the biases are responsible for the error correlations. To remove the possible influence of bias, we computed the bias-corrected error. On each trial, we subtracted the observer bias at each groundtruth tilt $e^* = \overbrace{(\hat{\tau} - \tau)}^{error} - \overbrace{E(\hat{\tau} - \tau|\tau)}^{bias}$ from the raw error. Human and model bias-corrected errors are also significantly correlated (*Figure 6C,D*). The human-human correlation (dashed line in *Figure 6B,D*; see *Figure 6—figure supplement 1*) sets an upper bound for the model-human correlation. The model-human correlation approaches this bound in some cases. Other measures of trial-by-trial similarity (e.g., choice probability; *Figure 6—figure supplement 2C*) yield similar conclusions. These results show that natural stimulus variation at a given groundtruth tilt causes similar response variation in human observers and the model observer.

To ensure that the predictive power of the model observer is not trivial, we developed multiple alternative models. All other models predict human performance more poorly (*Figure 6—figure supplement 2*). Our results do not rule out the possibility that another model could predict human performance better, but the current MMSE estimator establishes a strong benchmark against which other models must be compared.

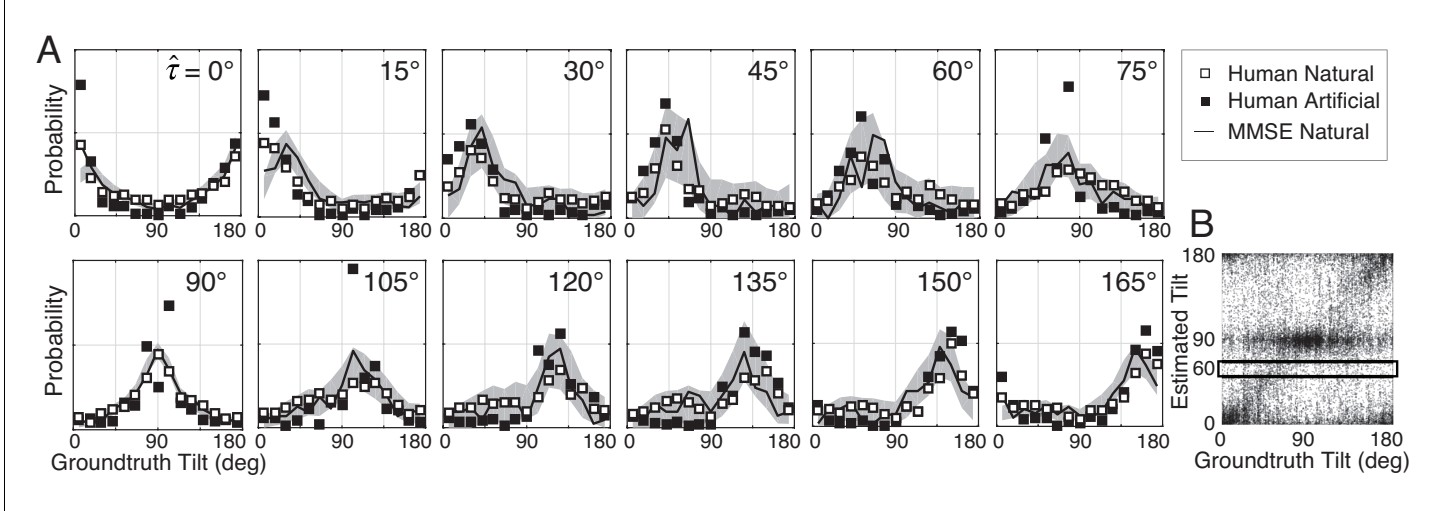

**Figure 4.** Distribution of groundtruth tilts for different tilt estimates. (A) Conditional distributions of groundtruth tilt $p(\tau|\hat{\tau})$ are obtained by binning groundtruth tilts for each estimated tilt (horizontal bins in *Figure 4B*). Unlike the conditional error distributions, these distributions are similar with natural and artificial stimuli. The most probable groundtruth tilt, conditional on the estimate, peaks at the estimated tilt for both stimulus types. Thus, any given estimate is a good indicator of the groundtruth tilt despite the overall poorer performance with natural stimuli. Also, these conditional distributions are well accounted for by the MMSE estimates; shaded regions show 95% confidence intervals on the MMSE estimates from 1000 Monte Carlo simulations of the experiment (see Materials and methods). The MMSE model had zero free parameters to fit to human performance. (B) Raw unsigned tilt estimates (same data as *Figure 2C*, but shown in the unsigned tilt domain). The box shows groundtruth tilts in the $\tau = 60°$ estimated tilt bin.

DOI: https://doi.org/10.7554/eLife.31448.011

The following figure supplement is available for figure 4:

**Figure supplement 1.** Alternative visualization of data in *Figures 3* and *4*.
DOI: https://doi.org/10.7554/eLife.31448.012

Thus, the normative model, without fitting to the human data, accounts for human tilt estimates at the level of the summary statistics (*Figure 2D–F*), the conditional distributions (*Figure 3* and *Figure 4*), and the trial-by-trial errors (*Figure 6*). Together, this evidence suggests that the human visual system's perceptual processes and the normative model's computations are making similar use of similar information. We conclude that the human visual system makes near-optimal use of the available information in natural stimuli for estimating 3D surface tilt.

## Performance-impacting stimulus factors: Slant, distance, and natural depth variation

In our experiment, natural and artificial stimuli were matched on many dimensions: tilt, slant, distance, and luminance contrast. These stimulus factors are commonly controlled in perceptual experiments. Consistent with previous reports, slant and distance had a substantial impact on estimation error (*Watt et al., 2005*) with both natural and artificial stimuli (*Figure 7*). (Luminance contrast had little impact on performance.)

Even after controlling for these stimulus dimensions, tilt estimation with natural stimuli is considerably poorer than tilt estimation with artificial stimuli. Other factors must therefore account for the differences. What are they? In our experiment, each artificial scene consisted of a single planar surface. Natural scenes contain natural depth variation (i.e., complex surface structure); some surfaces are approximately planar, some are curved or bumpy. How are differences in surface planarity related to differences in performance with natural and artificial scenes? To quantify the departure of surface structure from planarity, we defined local *tilt variance* as the circular variance of the groundtruth tilt values in the central 1° area of each stimulus. Then, we examined how estimation error changes with tilt variance.

First, we found that estimation error increases linearly with tilt variance for both human and model observers (*Figure 8A*). Unfortunately, tilt variance co-varies with groundtruth tilt — cardinal tilts

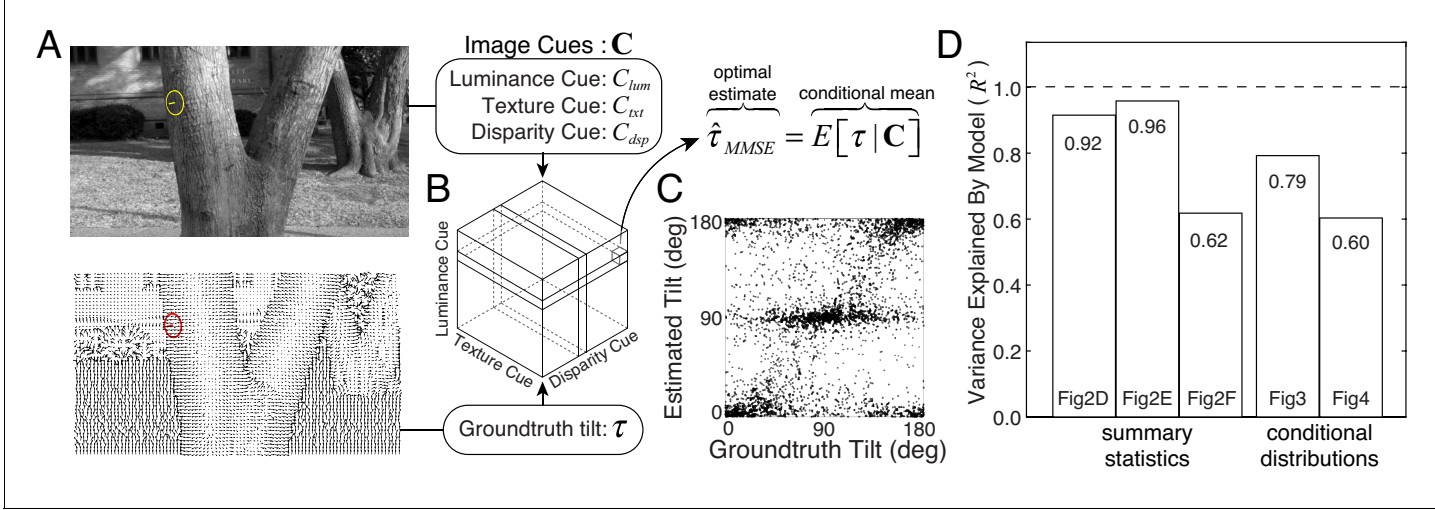

**Figure 5.** Normative model for tilt estimation in natural scenes. (**A**) The model observer estimate is the minimum mean squared error (MMSE) tilt estimate $\hat{\tau}_{MMSE}$ given three image cue measurements. Optimal estimates are approximated from 600 million data points (90 stereo-images) in the natural scene database: image cue values are computed directly from the photographic images and groundtruth tilts are computed directly from the distance data. (**B**) MMSE estimates for ~260,000 ($64^3$) unique image cue triplets are stored in an 'estimate cube.' (**C**) Model observer estimates for the 3600 unique natural stimuli used in the experiment. For each stimulus used in the experiment, the image cues are computed, and the MMSE estimate is looked up in the 'estimate cube.' Excluding the 3600 experimental stimuli from the 600 million stimuli that determined the estimate cube has no impact on predictions. The optimal estimates within the estimate cube change smoothly with the image cue values; hence, a relatively small number of samples can explore the structure of the full 3D space and provide representative performance measures (see Discussion). (**D**) Proportion variance explained ($R^2$) by the normative model for the summary statistics (estimate counts, means, and variances; *Figure 2D–F*) and the conditional distributions (*Figures 3* and *4*). All $R^2$ values are highly significant ($p<10^{-6}$).

DOI: https://doi.org/10.7554/eLife.31448.013

The following figure supplement is available for figure 5:

**Figure supplement 1.** Unsigned tilt estimates: human observers and normative model.

DOI: https://doi.org/10.7554/eLife.31448.014

tend to have lower tilt variance than oblique tilts, presumably because of the ground plane (*Figure 8B*)— which means that the effect of groundtruth tilt could be misattributed to tilt variance. Hence, we repeated the analysis of overall error separately for cardinal tilts alone and for oblique tilts alone. We found that the effect of tilt variance is independent of groundtruth tilt (*Figure 8C*). Thus, like slant and distance, tilt variance (i.e., departure from surface planarity) is one of several key stimulus factors that impacts tilt estimation performance.

Second, we found that for near-planar natural stimuli, average estimation error with natural and artificial stimuli are closely matched (left-most points in *Figure 8A*). Does this result mean that tilt variance accounts for *all* performance differences between natural and artificial stimuli? No. Performance with near-planar natural stimuli is still substantially different from performance with artificial stimuli (*Figure 8—figure supplement 1*). In addition, individual human and model trial-by-trial estimation errors are still correlated for the near-planar natural stimuli. Furthermore, the patterns of human performance with natural stimuli are robust across a wide range of tilt variance. *Figure 9* shows the summary statistics (estimate counts, means, and variances; cf. *Figure 2D–F*) for multiple different tilt variances of human observers. Model performance is also similarly robust to tilt variance (*Figure 9—figure supplement 1*).

We conclude that although tilt variance is an important performance-modulating factor, it is not the only factor responsible for performance differences with natural and artificial stimuli. Other factors must be responsible. Understanding these other factors is an important direction for future work.

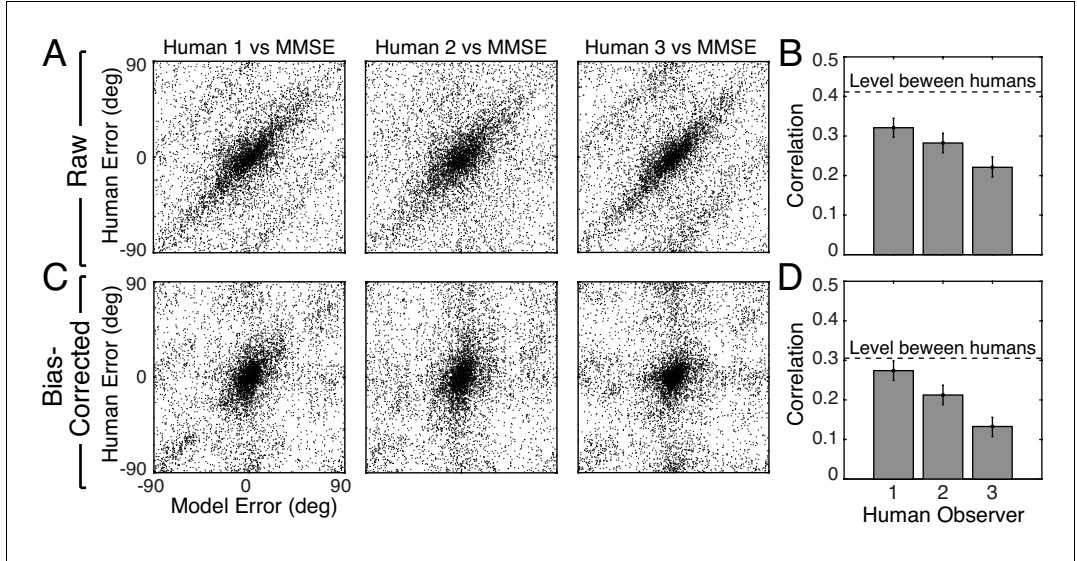

**Figure 6.** Trial-by-trial estimation errors: normative model vs. human observers. The diagonal structure in the plots indicates that trial-by-trial errors are correlated. (A) Raw trial-by-trial errors with natural stimuli between model and human observers. (B) Correlation coefficients (circular) for trial-by-trial errors between model and each human observer. The error bars represent 95% confidence intervals from 1000 bootstrapped samples of the correlation coefficient. The dashed line shows the mean of the correlation coefficients of errors between human observers in natural stimuli (*Figure 6—figure supplement 1*). (C) Bias-corrected errors in natural stimuli. (D) Correlation coefficient for bias-corrected errors.

DOI: https://doi.org/10.7554/eLife.31448.015

The following figure supplements are available for figure 6:

**Figure supplement 1.** Trial-by-trial estimation errors between humans.
DOI: https://doi.org/10.7554/eLife.31448.016

**Figure supplement 2.** Six alternative models for predicting human tilt estimation performance.
DOI: https://doi.org/10.7554/eLife.31448.017

## Discussion

Estimating 3D surface orientation requires the estimation of both slant and tilt. The current study focuses on tilt estimation. We quantify performance in natural scenes and report that human tilt estimates are often neither accurate nor precise. To connect our work to the classic literature, we matched artificial and natural stimuli on the stimulus dimensions that are controlled most often in typical experiments. The comparison revealed systematic performance differences. The detailed patterns of human performance are predicted, without free parameters to fit the data, by a normative model that is grounded in natural scene statistics and that makes the best possible use of the available image information. Importantly, this model is distinguished from many models of mid-level visual tasks because it is 'image computable'; that is, it takes image pixels as input and produces tilt estimates as output. Together, the current experiment and modeling effort contributes to a broad goal in vision and visual neuroscience research: to generalize our understanding of human vision from the lab to the real world.

### Generality of conclusions and future directions
#### Influence of full-field viewing
The main experiment examined tilt estimation performance for small patches of 3D natural scenes (1° and 3° of visual angle). Does tilt estimation performance improve substantially with full-field viewing of the 3D natural scenes? We re-ran the experiment with full-field viewing (36° x 21°; see *Figure 1B* for an example full-field scene). We found that human performance is essentially the same (*Figure 2—figure supplement 5*). Although it may seem surprising that full-field viewing does not substantially improve performance, it makes sense. Scene structure is correlated only over a local

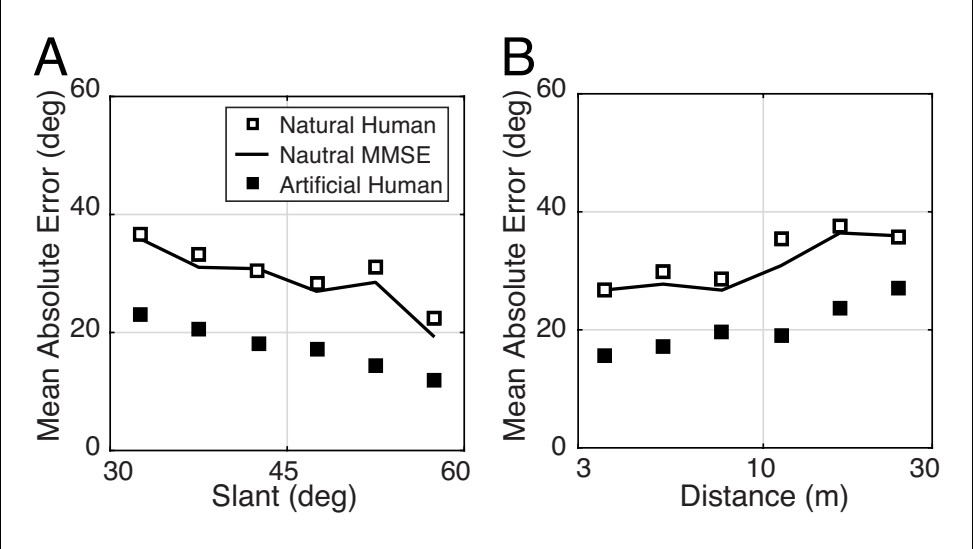

**Figure 7.** The effect of slant and distance on tilt estimation error in natural stimuli for human and model observers. (**A**) Absolute error decreases linearly with slant. Estimation error decreases approximately 20° as slant changes from 30° to 60°. (**B**) Absolute error increases linearly with distance. Estimation error increases approximately 15° as distance increases from 3 m to 30 m.

DOI: https://doi.org/10.7554/eLife.31448.018

area. Except for the ground plane, it is unusual for surfaces to have constant orientation over large visual angles. Thus, scene locations far from the target add little information about local tilt.

## Influence of scale

Groundtruth surface orientation is computed from a locally planar approximation to the surface structure, but surfaces in natural scenes are generally non-planar. Hence, the area over which groundtruth tilt is computed can affect the values assigned to each surface location. The same is true of the local image cue values. We checked how sensitive our results are to the scale of the local analysis area. We recomputed groundtruth tilt for two scales and recomputed image cue values for four scales (see Materials and methods). All eight combinations of scales yield the same qualitative pattern of results.

## Influence of gaze angle

The statistics of local surface orientation change with elevation in natural scenes (*Adams et al., 2016*; *Yang and Purves, 2003b*). In our study, scene statistics were computed from range scans and stereo-images (36° x 21° field-of-view) that were captured from human eye height with earth parallel gaze (*Burge et al., 2016*). Different results may characterize other viewing situations, a possibility that could be evaluated in future work. However, the vast majority of eye movements in natural scenes are smaller than 10° (*Land and Hayhoe, 2001*; *Pelz and Rothkopf, 2007*; *Dorr et al., 2010*). Hence, the results presented here are likely to be representative of an important subset of conditions that occur in natural viewing.

## Influence of internal noise

We examined how well the normative model (i.e., MMSE estimator) predicts human performance with artificial stimuli. The model nicely predicts the unbiased pattern of human estimate means. However, the model predicts estimate variances that are lower than the human estimate variances that we observed (although the predicted and observed patterns are consistent). We do not yet understand the reason for this discrepancy. One possibility is that the normative model used here does not explicitly model how internal noise affects human performance. In natural scenes, natural stimulus variability may swamp internal noise and be the controlling source of uncertainty. But with

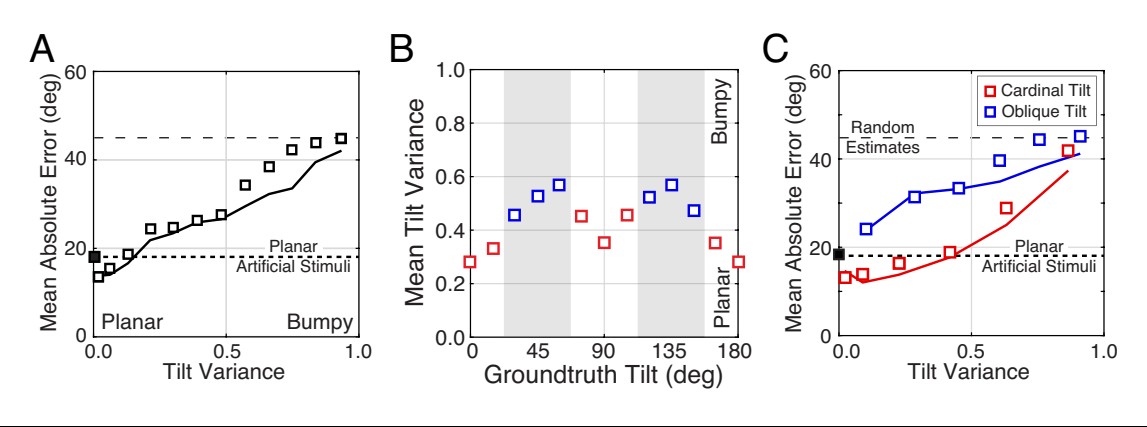

**Figure 8.** The effect of tilt variance on tilt estimation error. (**A**) Absolute error increases linearly with tilt variance. Estimation error increases approximately 25° across the range of tilt variance. Artificial stimuli were perfectly planar and had zero local depth variation; hence the individual data point at zero tilt variance. Solid curve shows the model prediction. (**B**) Tilt variance co-varies with groundtruth tilt. Oblique tilts tend to be associated with less planar (i.e., more bumpy) regions of natural scenes. (Tilt variance was computed in 15° wide bins.) (**C**) Same as (**A**) but conditional on whether groundtruth tilts are cardinal (red, 0° ± 22.5° or 90° ± 22.5°) or oblique (blue, 45° ± 22.5° or 135° ± 22.5°, shaded areas in [**B**]). Data points are spaced unevenly because they are grouped in quantile bins, such that each data point represents an equal number of stimuli. The solid curves represent the errors of the MMSE estimator for cardinal (red) and oblique (blue) groundtruth tilts. The normative model predicts performance in all cases.
DOI: https://doi.org/10.7554/eLife.31448.019

The following figure supplement is available for figure 8:

**Figure supplement 1.** Tilt estimation performance with near-planar natural stimuli .
DOI: https://doi.org/10.7554/eLife.31448.020

artificial stimuli, an explicit model of internal noise may be required to account quantitatively for the variance of human performance. Determining the relative importance of natural stimulus variability and internal noise is an important topic for future work.

## Influence of sampling error

The natural stimuli presented in the experiment were chosen via constrained random sampling (see Materials and methods). Random stimulus sampling increases the likelihood that the reported performance levels are representative of generic natural scenes. One potential concern is that the relatively small number of unique stimuli that can be practically used in an experiment (e.g., n = 3600 in this experiment) precludes a full exploration of the space of optimal estimates (see *Figure 5B*). Fortunately, the tilt estimates from the normative model change smoothly with image cue values. Systematic sparse sampling should thus be sufficient to explore the space. To rigorously determine the influence of each cue on performance, future parametric studies should focus on the role of particular image cue combinations and other important stimulus dimensions such as tilt variance.

## Influence of non-optimal cues

Although the three local image cues used by the normative model are widely studied and commonly manipulated, there is no guarantee that they are the most informative cues in natural scenes. Automatic techniques could be used to find the most informative cues for the task (*Geisler et al., 2009*; *Burge and Jaini, 2017*;*Jaini and Burge, 2017*). These techniques have proven useful for other visual estimation tasks with natural stimuli (*Burge and Geisler, 2011Burge and Geisler, 2012, 2014, 2015*). However, in the current task, we speculate that different local cues are unlikely to yield substantially better performance (*Burge et al., 2016*). Also, given the similarities between human and model observer performance, any improved ability to predict human performance is likely to be modest at best. Nevertheless, the only way to be certain is to check.

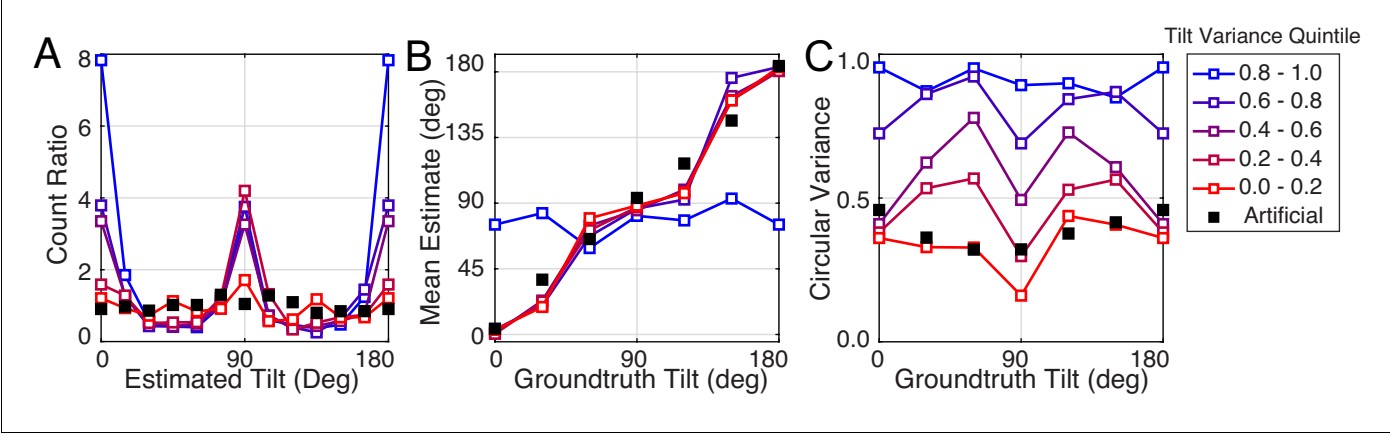

**Figure 9.** Robustness of performance measures to tilt variance. Human tilt estimation performance with natural stimuli for five tilt variance quintiles (colors). The quintile centers are at 0.12, 0.33, 0.55, 0.76, and 0.97, respectively. (**A**) Estimate count ratio (i.e., the ratio of estimated to presented tilt) at each tilt. With near-planar natural stimuli, cardinal tilts are still estimated much more frequently than with planar artificial stimuli. (**B**) Estimate means. (**C**) The variance of estimates. Except with the highest tilt variance stimuli, the patterns of mean and variance with natural stimuli hold across tilt variances, except for natural stimuli with the highest tilt variance.

DOI: https://doi.org/10.7554/eLife.31448.021

The following figure supplement is available for figure 9:

**Figure supplement 1.** Tilt estimation performance of the normative model with natural stimuli for five tilt variance quintiles.

DOI: https://doi.org/10.7554/eLife.31448.022

## 3D surface orientation estimation

The estimation of the 3D structure of the environment is aided by the joint estimation of tilt and slant (Marr's '2.5D sketch') (**Marr, 1982**). Although we have shown that human and model tilt estimation performance are systematically affected by surface slant (**Figure 7A**), the current work only addresses the human ability to estimate unsigned tilt. We have not yet explicitly modeled how humans estimate signed tilt, how humans estimate slant, or how humans jointly estimate slant and tilt. We will attack these problems in the future.

## Cue-combination with and without independence assumptions

The standard approach to modeling cue-combination, sometimes known as maximum likelihood estimation, includes a number of assumptions: a squared error cost function, cue independence, unbiased Gaussian-distributed single cue estimates, and a flat or uninformative prior (**Ernst and Banks, 2002**) (but see [**Oruç et al., 2003**]). The approach used here (normative model; see **Figure 5**) assumes only a squared error cost function, and is guaranteed to produce the Bayes optimal estimate given the image cues, regardless of the common assumptions . In natural scenes, it is often unclear whether the common assumptions hold. Methods with relatively few assumptions can therefore be powerful tools for establishing principled predictions. We have not yet fully investigated how the image cues are combined in tilt estimation, but we have conducted some preliminarily analyses. For example, a simple average of the single-cue estimates (each based on luminance, texture, or disparity alone) underperforms the three-cue normative model. This result is not surprising given that the individual cues are not independent, that the single cue estimates do not follow Gaussian distribution, and that the tilt prior is not flat. However, the current study is not specifically designed to examine the details of cue combination in tilt estimation. To examine cue-combination in this task rigorously, a parametric stimulus-sampling paradigm should be employed, a topic that will be explored in future work.

## Local and global tilt estimation

A grand problem in perception and neuroscience research is to understand how local estimates are grouped into more accurate global estimates. We showed that local tilt estimates are unbiased predictors of groundtruth tilt and have nearly equal reliability (**Figure 4**). This result implies that optimal

spatial pooling of the local estimates may be relatively simple. Assuming statistical independence (i. e., naïve Bayes), optimal spatial pooling is identical to a simple linear combination of the local estimates: the straight average of $N$ local estimates $\hat{\tau}^{global} = \frac{1}{N}\sum_i^N \hat{\tau}_i^{local}$. Of course, local groundtruth tilts and estimates are spatially correlated, so the independence assumption will not be strictly correct. However, the spatial correlations could be estimated from the database and incorporated into the computations. Our work thus lays a strong empirically grounded foundation for the investigation of local-global processing in surface orientation estimation.

## Behavioral experiments with natural images

In classic studies of surface orientation perception, stimuli are usually limited in at least one of two important respects. If the stimuli are artificial (e.g., computer-graphics generated), groundtruth surface orientation is known but lighting conditions and textures are artificial, and it is uncertain whether results obtained with artificial stimuli will generalize to natural stimuli. If the stimuli are natural (e.g., photographs of real scenes), groundtruth surface orientation is typically unknown which complicates the evaluation of the results. The experiments reported here used natural stereo-images with laser-based measurements of groundtruth surface orientation, and artificial stimuli with tilt, slant, distance, and contrast matched to the natural stimuli. This novel design allows us to relate our results to the classic literature, to determine the generality of results with both natural and artificial stimuli and to isolate performance-controlling differences between the stimuli. In particular, we found that tilt variance is a pervasive performance-altering feature of natural scenes that is not explicitly considered in most investigations. The human visual system must nevertheless contend with tilt variance in natural viewing. We speculate that characterizing its impact is likely to be fundamental for understanding 3D surface orientation estimation in the real-world, just as characterizing the impact of local luminance contrast has been important for understanding how humans detect spatial patterns in noise (*Burgess et al., 1981*).

## Perception and the internalization of natural scene statistics

The current study is the latest in a series of reports that have attempted, with ever increasing rigor, to link properties of perception to the statistics of natural images and scenes. Our contribution extends previous work in several respects. First, previous work demonstrated similarity between human and model performance only at the level of summary statistics (*Girshick et al., 2011*; *Burge et al., 2010b*; *Weiss et al., 2002*; *Stocker and Simoncelli, 2006*). We demonstrate that a principled model, operating directly on image data, predicts the summary statistics, the distribution of estimates, and the trial-by-trial errors. Second, previous work showed that human observers behave as if their visual systems have encoded the task-relevant statistics of 2D natural images (*Girshick et al., 2011*). We show that human observers behave as if they have properly encoded the task-relevant joint statistics of 2D natural images and the 3D properties of natural scenes (also see (*Burge et al., 2010b*)). Third, previous work tested and modeled human performance with artificial stimuli only (*Girshick et al., 2011*; *Burge et al., 2010b*; *Weiss et al., 2002*; *Stocker and Simoncelli, 2006*). We test human performance with both natural and artificial stimuli. The dramatic, but lawful, changes in performance with natural stimuli highlight the importance of studies with the stimuli that visual systems evolved to process.

# Materials and methods

## Apparatus

The stereo images were presented with a ViewPixx Technologies ProPixx projector fitted with a 3D polarization filter. Left and right images were presented sequentially at a refresh rate of 120 Hz (60 Hz per eye) and with the same resolution of the two images (1920 × 1080 pixel). The observer was positioned 3.0 m from a 2.0 × 1.2 m Harkness Clarus 140 XC polarization maintaining projection screen. This viewing distance minimizes the potential influence of screen cues to flatness (e.g., blur). Human observers wore glasses with passive (linear) polarized filters to isolate the image for the left and right eyes. The observer's head was stabilized with a chin- and forehead-rest. From this viewing position, the projection screen subtended 36° x 21° of visual angle. The disparity-specified distance created by this projection system matched to the distances measured in the original natural scenes.

The projection display was linearized over 10 bits of gray level. The maximum luminance was 84 cd/m$^2$. The mean luminance was set to 40% of the projection system's maximum luminance.

## Participants

Three human observers participated in the experiment; two were authors, and one was naïve about the purpose of the experiment. Informed consent was obtained from participants before the experiment. The research protocol was approved by the Institutional Review Board of the University of Pennsylvania and is in accordance with the Declaration of Helsinki.

## Experiment

Human observers binocularly viewed a small region of a natural scene through a circular aperture (1° or 3° diameter) positioned 5 arcmin of disparity in front of the scene point along the cyclopean line of sight. Observers communicated their tilt estimate with a mouse-controlled probe. Each observer viewed 3600 unique natural stimuli (150 stimuli per tilt bin x 24 tilt bins) presented with each of two apertures in the experiment (7200 total). Natural stimuli were constrained to be binocularly visible (no half-occlusions), to have slants larger than 30°, to have distances between 5 m and 50 m, and to have contrasts between 5% and 40%. Each observer also viewed 1440 unique artificial stimuli (60 stimuli per tilt bin x 24 tilt bins) with two apertures (2880 total). Artificial stimuli (1/f noise and phase- and orientation-randomized plaids) were matched to the natural stimuli on multiple additional dimensions (tilt, slant, distance, and contrast). Natural stimuli were presented in 48 blocks of 150 trials each, and artificial stimuli were presented in 12 blocks of 240 trials each, with interleaved blocks using small and large apertures.

## Data analysis

Tilt is a circular (angular) variable. We computed the mean, variance, and error using standard circular statistics. The circular mean is defined as $\bar{\tau} = arg[\mathbf{R}]$ where $\mathbf{R} = \left[\sum_\tau exp[j\tau]\right]/N$ is the complex mean resultant vector. The circular variance is defined as $var(\tau) = 1 - |\mathbf{R}|$. Estimation error $\mathbf{e} = arg[exp[j(\hat{\tau} - \tau)]]$ is the circular distance between the tilt estimate and groundtruth.

## Groundtruth tilt

Groundtruth tilt $\tau$ is computed from the distance data (range map $\mathbf{r}$) co-registered to each natural image in the database. We defined groundtruth tilt $tan^{-1}(\nabla_y r/\nabla_x r)$ as the orientation of the normalized range gradient (*Marr, 1982*). The range gradient was computed by convolving the distance data with a 2D Gaussian kernel having space constant $\sigma$ and then taking the partial derivatives in the $x$ and $y$ image directions (*Burge et al., 2016*). For the results presented in this manuscript, groundtruth tilt was computed using a space constant of $\sigma = 3$ arcmin; doubling this space constant does not change the qualitative results. The space constants correspond to kernel sizes of ~0.25°−0.50°.

## Image cues to tilt

Image cues to tilt (disparity, luminance, and texture cues) were computed directly from the images. Like groundtruth tilt, image cues were defined as the orientation $tan^{-1}(\nabla_y cue/\nabla_x cue)$ of the local disparity and luminance gradients. The local disparity gradient is computed from the disparity image, which is obtained from the left and right eye luminance images via standard local windowed cross-correlation (*Burge et al., 2016*; *Tyler and Julesz, 1978*; *Banks et al., 2004*). The window for cross-correlation had the same space constant as the derivative operator that was used to compute the gradient (see below). The texture cue to tilt is defined as the orientation of the major axis of the local amplitude spectrum of the luminance image. This texture cue is non-standard (but see [*Fleming et al., 2011*]). However, this texture cue is more accurate in natural scenes than traditional texture cues (*Burge et al., 2016*; *Clerc and Mallat, 2002*; *Galasso and Lasenby, 2007*; *Malik and Rosenholtz, 1997*; *Massot and Hérault, 2008*). For the main results presented in this manuscript, image cues were computed from the gradients using a space constant of $\sigma = 6$ arcmin; using the space constants to $\sigma = 3$, 6, 9, or 12 arcmin does not change the qualitative results. The space constants correspond to kernel sizes of ~0.25°−1.0°.

## Local luminance contrast

Luminance contrast was defined as the root-mean-squared luminance values within a local area weighted by a cosine window. Specifically, luminance contrast is $C = \sqrt{\left[\sum_{\mathbf{x}\in A} ((I(\mathbf{x}) - \bar{I})/\bar{I})^2 W(\mathbf{x})\right]/\sum_{\mathbf{x}\in A} W(\mathbf{x})}$ where $\mathbf{x}$ is the spatial location, $W$ is a cosine window with area $A$, and $\bar{I} = \left[\sum_{\mathbf{x}\in A} I(\mathbf{x})W(\mathbf{x})\right]/\sum_{\mathbf{x}\in A} W(\mathbf{x})$ is the local mean intensity.

## The output-mapping problem

On each trial, human observers communicated their perceptual estimate $\hat{\tau}$ by making a response $\hat{\tau}_{rsp}$ with a mouse-controlled probe. Unfortunately, the responses are not guaranteed to equal the perceptual estimates. An output-mapping function $\hat{\tau}_{rsp} = g(\hat{\tau})$ relates the response to the perceptual estimate, and an estimation function $\hat{\tau} = f(\tau)$ relates the estimate to the groundtruth tilt of each stimulus. When responses are biased, it is hard to conclude whether the biases are due to the output-mapping function or to the estimation function. When responses are unbiased, a stronger case can be made that the human responses equal the perceptual estimates. To obtain unbiased responses $\hat{\tau}_{rsp} = \tau$ from biased estimates $\hat{\tau} \neq \tau$, the output mapping function would have to equal exactly the inverse of a biased estimation function: $g(.) = f^{-1}(.)$; this possibility seems unlikely and has no explanatory power. Thus, by Occam's razor, unbiased responses imply unbiased output-mapping and estimation functions: $\hat{\tau}_{rsp} = \hat{\tau} = \tau$. Human responses to artificial stimuli were unbiased (*Figure 2E*), implying an unbiased output-mapping function. Assuming that the output-mapping function is stable across stimulus types, we conclude that the biased responses to natural stimuli accurately reflect biased perceptual estimates.

## Monte Carlo simulations

To determine whether the model predictions are representative of randomly sampled natural stimuli, we simulated 1000 repeats of the experiment. On each repeat, we obtained a different sample of 3600 natural stimuli (150 in each tilt bin) from which we obtained 3600 optimal estimates. The samples are used to compute 95% confidence intervals on the model predictions, which are shown as the shaded regions in *Figure 3A* and *Figure 4A*.

# Additional information

### Funding

| Funder | Grant reference number | Author |
| --- | --- | --- |
| National Institutes of Health | EY011747 | Johannes Burge |
| University of Pennsylvania | Startup Funds | Johannes Burge |

The funders had no role in study design, data collection and interpretation, or the decision to submit the work for publication.

### Author contributions

Seha Kim, Conceptualization, Resources, Data curation, Software, Formal analysis, Validation, Investigation, Visualization, Methodology, Writing—original draft, Project administration, Writing—review and editing; Johannes Burge, Conceptualization, Resources, Software, Formal analysis, Supervision, Funding acquisition, Validation, Investigation, Visualization, Methodology, Writing—original draft, Project administration, Writing—review and editing

### Author ORCIDs

Seha Kim (iD) http://orcid.org/0000-0003-0356-6168
Johannes Burge (iD) http://orcid.org/0000-0002-0311-7875

### Ethics

Human subjects: Informed consent was obtained from participants before the experiment. The research protocol was approved by the Institutional Review Board of the University of Pennsylvania (IRB approval protocol number: 824435) and is in accordance with the Declaration of Helsinki.

### Decision letter and Author response

Decision letter https://doi.org/10.7554/eLife.31448.029
Author response https://doi.org/10.7554/eLife.31448.030

## Additional files

### Supplementary files

• Source data 1. Source data for full-field natural scene experiment.
DOI: https://doi.org/10.7554/eLife.31448.023

• Source data 2. Source data for human performance.
DOI: https://doi.org/10.7554/eLife.31448.024

• Source data 3. Source data for MMSE model.
DOI: https://doi.org/10.7554/eLife.31448.025

• Transparent reporting form
DOI: https://doi.org/10.7554/eLife.31448.026

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
