## [Decision Letter]

Thank you for submitting your article "The lawful imprecision of human surface tilt estimation in natural scenes" for consideration by *eLife*. Your article has been reviewed by two peer reviewers, and the evaluation has been overseen by Reviewing Editor Jack Gallant, and David Van Essen as the Senior Editor. The following individuals involved in review of your submission have agreed to reveal their identity: Mark Lescroart (Reviewer #1); Michael Landy (Reviewer #2).

The reviewers have discussed the reviews with one another and the Reviewing Editor has drafted this decision to help you prepare a revised submission.

Summary:

The authors study the ability of human subjects to estimate surface tilt in natural images. They find (among other results) that humans are biased to estimate tilt at cardinal (vertical and horizontal) orientations and show that an image-computable Bayesian model makes estimates of tilt that are similar to human estimates. Both reviewers found the paper to be interesting and timely. Reviewer 2 made only minor comments, but reviewer 1 had some concerns and requested additional behavioral data. The reviewing editor is persuaded by these arguments.

Essential revisions:

1) The authors have found that variance (or noise) in tilt in the stimulus leads to less accurate estimation of tilt. However, the natural images used in this study are highly variable, so this result isn't all that surprising. The authors should analyze whether the model is predicting human performance well simply because some trials have more or less tilt variance than others. If this is the case, the result is much less interesting – variance (or noise) in a tilt *should* cause poorer tilt estimates. Similarly, alternative versions of the plots in Figure 3 should be generated with low-tilt-variance scenes, to see if the bias shows up as clearly.

2) The authors should validate their results using artificial images with more naturalistic textures (e.g., 1/f). In general, the authors should try to introduce variability into the artificial images of the same kind and magnitude as the variability in the natural images.

3) The authors should perform some control experiments to verify that the results hold for stimuli larger than 3 degrees. If possible, it would be good to verify these effects in complex natural scenes.

Reviewer #1:

The authors study the ability of human subjects to estimate surface tilt in natural images. They find (among other results) that humans are biased to estimate tilt at cardinal (vertical and horizontal) orientations and show that an image-computable Bayesian model makes estimates of tilt that are similar to human estimates.

This is an interesting and timely question, and the study is generally well executed. The figures are nice, the writing is clear, and the dataset the authors use (if it is to be shared) seems a useful contribution. However, I have some concerns about the experimental paradigm. None of these concerns alone are fatal flaws, but when combined with some of the results, they give me doubts about the impact of the rest of the results.

First, the artificial stimuli are perhaps too simplified (very regularly textured, wholly planar). This is not representative of studies of tilt estimation: several studies have had human subjects estimate surface orientation (or related quantities) of non-planar surfaces (e.g. Todd et al., 1996; Li and Zaidi, 2000; Norman et al., 2006). The highly simple nature of the artificial stimuli here creates several obvious differences between the natural and artificial images. Figure 8 shows that one such difference – tilt variance, present in the natural images but not in the artificial ones – accounts for all of the difference in mean tilt estimation accuracy between artificial and natural images. Stated another way, the authors have found that variance (or noise) in tilt in the stimulus leads to less accurate estimation of tilt. Note that this is not noise in the image cues or anything else – this is variance in the exact parameter that is being estimated. I may be missing something, but this particular result (which appears to be the biggest effect in the experiment) seems wholly expected to me. So, I am unimpressed by the conclusion statement: "The dramatic, but lawful, fall-off in performance with natural stimuli highlights the importance of performing studies with the stimuli visual systems evolved to process."

The large effect of tilt variance calls into question the size of other effects the authors report. Figure 8—figure supplement 1 shows that, for natural stimuli with low tilt variance, the bias toward estimating vertical (0 and 180 degree) tilts is greatly diminished (the count ratio between estimated and true instances of vertical tilt is very near to 1). (As a side note, Figure 8—figure supplement 1 is a critical figure and should appear in the main manuscript). It is also not clear how much tilt variance might be affecting the model's predictions of trial-to-trial errors; the authors should analyze whether the model is predicting human performance well simply because some trials have more or less tilt variance than others. If this is the case, the result is much less interesting – variance (or noise) in a tilt *should* cause poorer tilt estimates. Similarly, alternative versions of the plots in Figure 3 should be generated with low-tilt-variance scenes, to see if the bias shows up as clearly.

The other striking difference between the artificial and natural images is the extreme regularity of the textures in the artificial images (at least of the plaids shown in Figure 2). The authors also used 1/f noise as a texture in the artificial images – did human performance differ depending on whether the artificial stimuli were plaid or 1/f noise? In general, it seems that adding more types of variation to the artificial stimuli and assessing the effects of that variation would provide a good way to assess what sorts of variation make human performance look more like it does for natural images. I suspect that the authors plan to do this in future work, but I think it would substantially increase the impact of this work to include such data and analyses here.

Finally – I admit some disappointment with the choice of only showing 3 degree stimuli. To me, this lessens the impact of the work as well; a 3 degree image patch hardly constitutes a "scene". Thus, the following conclusion statement seems a bit of a reach: "We quantify performance in natural scenes and report that human tilt percepts are often neither accurate nor precise". Human estimates of tilt given full natural images (including much more context) would likely be better than the estimates reported here. I realize this is a very difficult problem, but *eLife* is also a broad, prestigous journal; studying tilt estimation in natural image *patches* may be a critical step on the way to studying tilt estimation in full scenes, but it also seems less broadly interesting.

Last, a few notes on the model: First, I am puzzled as to why the authors do not include model performance on their artificial stimuli, too. This seems to be a straightforward and easy test of the generality of the model. Second, it's not clear to me whether the "estimate cube" of optimal mappings between image cues and tilts is computed using some, all, or none of the same images that the subjects saw in the experiment. The authors should clarify this point.

I should note again that the concerns above are almost entirely about impact. I hesitate to reject an otherwise interesting and well-executed study on grounds that it's just not splashy enough. And there are several interesting and solid results in this paper. The fact that tilt variance is correlated with tilt angle in a large sample of natural scenes seems solidly supported and important. Modulo the questions I raised above, the MMSE model performance appears to provide a good match for human performance in the natural images. The persistent difference between errors estimating cardinal and oblique tilt, as well as the persistent bias to estimate horizontal tilt – both with matched tilt variance – are also interesting. Thus, I am on the fence about this paper, mostly because its impact seems marginal. I could be convinced to accept the paper with revisions or to reject the paper.

*Reviewer #2:*

This is a lovely paper, showing that a nonparametric Bayesian model of tilt estimation accounts startlingly well for human behavior in a tilt-estimation task. My comments are mainly about improving the clarity, not much more.

Introduction: Many of my comments are a result of reading it in (my) natural order, i.e., your page order with diversions to the Methods when needed. So, when I got here I wondered whether the patches to be judged were centered on the display or occluded in the position in the original images. That's never stated explicitly but implied by a figure that hasn't come up yet.

Introduction: You never motivate/justify pooling over tilt sign until much, much later, and so I was surprised you threw information away from the start. I wondered about it again for Figure 3 where, given that you provide disparity, the tilt sign ambiguity from pictorial cues should be alleviated.

Figure 2 et seq.: Why didn't you run the model on the artificial stimuli and show the model fits for those data points (or misfits, as the case may be)?

Subsection “Normative model”: The citation of Figure 6—figure supplement 2 here seems out of place. The analyses for this figure don't appear until the next page. Also, shouldn't all the supplementary figures be cited somewhere in the main text? I think a bunch aren't.

Subsection “Trial-by-trial Error: Is -> Are.

Figure 8:Exactly what bin cutoffs did you use for blue vs. red here?

Subsection “Effect of Natural Depth Variation”: artificially -> artificial.

Subsection “Generality of Conclusions and Future Directions”: our -> are.

Subsection “Cue-combination with and without independence assumptions”: This reference to Figure 6—figure supplement 2, since the pooled/averaged model is not in the figure, but merely mentioned in its legend.

Subsectiion “Experiment”: More details please: What's your definition of contrast, refer to the figure to state what part of the patch they were supposed to judge, were the judged bins over 180 or 360 degrees (I only say this because Figure 1 leads the reader to believe that it's over 180 degrees only). Please show and give details about the two types of artificial stimuli. Do responses to them differ from one another?

Subsection “Groundtruth tilt”: "atan2" is MATLAB notation, I'd think. You might want to say what you mean there.

Subsection “Image cues to tilt”: I'd like more detail here as well. The disparity cue must be based on a definition of "local" and a restriction of cross-correlation shifts. The disparity gradient requires a scale. The disparity gradient doesn't have a tilt sign ambiguity, but the texture cue does. I'm not sure "patch size at half height" won't confuse people (you don't mean the viewed patch, but rather the patch after multiplying by the Gaussian window).

Figure 6—figure supplement 2: Actually, I'm rather surprised that the single-cue models (especially those other than disparity) perform as well as they do. It worries me that there are weird regularities in your database. You never say what your definition of luminance is exactly, but why should tilt be dependent on luminance? How are each cue binned? Are the same bins used for the single-cue models, so that those models have vastly smaller measured parameters?

Figure 6—figure supplement 2 legend, line 12: "…but better than the prior or…".

---

## [Author Response]

Essential revisions:1) The authors have found that variance (or noise) in tilt in the stimulus leads to less accurate estimation of tilt. However, the natural images used in this study are highly variable, so this result isn't all that surprising. The authors should analyze whether the model is predicting human performance well simply because some trials have more or less tilt variance than others.

We agree that this is an important issue. The estimate means and variances as a function of groundtruth tilt are largely robust to changes in tilt variance (see new Figure 9). We also now show that the model does predict the distribution of human errors with near-planar natural stimuli (new addition to Figure 8—figure supplement 1). These results indicate that model’s success at predicting performance is not due simply to the fact that natural stimuli had higher tilt variance on average.

If this is the case, the result is much less interesting – variance (or noise) in a tilt should cause poorer tilt estimates.

We agree that if our model predicted nothing other than an increase in overall estimate variance, the result would not be particularly interesting. However, the model predicts the pattern of estimate means and variances as a function of groundtruth tilt and the distributions of tilt errors. These are all non-trivial predictions.

Similarly, alternative versions of the plots in Figure 3 should be generated with low-tilt-variance scenes, to see if the bias shows up as clearly.

We have done so. We analyzed the subset of near-planar natural stimuli in the experimental dataset and present the results in Figure 8—figure supplement 1. With low-tilt variance (i.e. near-planar) locations in natural scenes (i) the bias persists, (ii) human performance continues to be substantially different with natural and artificial stimuli, and (iii) human performance continues to be predicted by the model.

2) The authors should validate their results using artificial images with more naturalistic textures (e.g., 1/f). In general, the authors should try to introduce variability into the artificial images of the same kind and magnitude as the variability in the natural images.

We now present results (Figure 2—figure supplement 3) for the 1/f noise textures and two plaid textures separately (see below). The three textures yield similar results although the 1/f textures yield estimates with slightly higher variance. The main point to take from these plots is that the planar artificial stimuli produce very different patterns of results from the natural stimuli (e.g., the lack of bias and the different pattern of variances). Performance with these artificial stimuli is notably different than performance with the near-planar subset of natural stimuli.

3) The authors should perform some control experiments to verify that the results hold for stimuli larger than 3 degrees. If possible, it would be good to verify these effects in complex natural scenes.

As requested, we re-ran our experiment without apertures so that human observers had full-field views of each scene (36ºx21º). We added new Figure 2—figure supplement 5 that shows the estimate counts, means, variances, and conditional distributions for full-field viewing. The data with full-field viewing verifies that the results in the original experiment hold for stimuli larger than 3 degrees.

Reviewer #1:The authors study the ability of human subjects to estimate surface tilt in natural images. They find (among other results) that humans are biased to estimate tilt at cardinal (vertical and horizontal) orientations and show that an image-computable Bayesian model makes estimates of tilt that are similar to human estimates.This is an interesting and timely question, and the study is generally well executed. The figures are nice, the writing is clear, and the dataset the authors use (if it is to be shared) seems a useful contribution. However, I have some concerns about the experimental paradigm. None of these concerns alone are fatal flaws, but when combined with some of the results, they give me doubts about the impact of the rest of the results.First, the artificial stimuli are perhaps too simplified (very regularly textured, wholly planar). This is not representative of studies of tilt estimation: several studies have had human subjects estimate surface orientation (or related quantities) of non-planar surfaces (e.g. Todd et al., 1996; Li and Zaidi, 2000; Norman et al., 2006).

We agree that not all studies have been performed with planar surfaces, but most have. We have clarified the writing to make this point more clear and we now cite Norman et al. (2006), an unfortunate omission in our original submission.

The highly simple nature of the artificial stimuli here creates several obvious differences between the natural and artificial images. Figure 8 shows that one such difference – tilt variance, present in the natural images but not in the artificial ones – accounts for all of the difference in mean tilt estimation accuracy between artificial and natural images. Stated another way, the authors have found that variance (or noise) in tilt in the stimulus leads to less accurate estimation of tilt. Note that this is not noise in the image cues or anything else – this is variance in the exact parameter that is being estimated. I may be missing something, but this particular result (which appears to be the biggest effect in the experiment) seems wholly expected to me. So, I am unimpressed by the conclusion statement: "The dramatic, but lawful, fall-off in performance with natural stimuli highlights the importance of performing studies with the stimuli visual systems evolved to process." The large effect of tilt variance calls into question the size of other effects the authors report.

It is true that the absolute error with natural stimuli is smaller when there is no tilt variance; and we agree that this effect is not particularly surprising. But even when natural stimuli have low tilt variance (i.e., are near-planar), performance with natural and artificial stimuli is not the same. Thus, tilt variance alone cannot explain all differences in human performance with natural and artificial stimuli. See new Figure 9 and modified Figure 8—figure supplement 1.

That being said, tilt variance is a pervasive performance impacting stimulus factor in natural scenes that has not been systematically characterized before, and we think it important to do so. We speculate that characterizing the impact of tilt variance is likely to be fundamental understanding 3D surface orientation estimation in the real-world, just as characterizing the impact of local luminance contrast has been important for understanding how humans detect spatial patterns in noisy uncertain backgrounds (Burgess et al., 1981).

We have reorganized the text to make more clear exactly what we are and what we are not claiming. We hope our efforts to improve clarity make for easier reading.

Figure 8—figure supplement 1 shows that, for natural stimuli with low tilt variance, the bias toward estimating vertical (0 and 180 degree) tilts is greatly diminished (the count ratio between estimated and true instances of vertical tilt is very near to 1). (As a side note, Figure 8—figure supplement 1 is a critical figure and should appear in the main manuscript). It is also not clear how much tilt variance might be affecting the model's predictions of trial-to-trial errors; the authors should analyze whether the model is predicting human performance well simply because some trials have more or less tilt variance than others. If this is the case, the result is much less interesting – variance (or noise) in a tilt should cause poorer tilt estimates. Similarly, alternative versions of the plots in Figure 3 should be generated with low-tilt-variance scenes, to see if the bias shows up as clearly.

The count ratio varies between ~0.5 and ~2.0 for low-tilt-variance natural stimuli (Figure 8—figure supplement 1). Across all natural stimuli, the count ratio varied between ~0.5 and ~3.0. Also, the pattern of estimate bias persists with low-tilt-variance stimuli (Figure 8—figure supplement 1). More importantly, we now show the distributions of estimation error for natural stimuli with low tilt variance (Figure 8—figure supplement 1). There remain substantial differences in human performance with tilt-variance-matched natural and artificial stimuli.

All three human observers show significant trial-by-trial raw error correlations with the model for near-planar natural stimuli. Two of three observers show significant trial-by-trial bias corrected error correlations with the model for near-planar natural stimuli.

We have substantially re-written the Results section “Performance-impacting Stimulus Factors: Slant, Distance, & Natural Depth Variation” to make all these points more clear. We have included a new Figure 9, which shows that, although there is an effect of tilt variance, the basic performance patterns are robust to changes in tilt variance. We respectfully decline to move Figure 8—figure supplement 1nto the main text as we think it breaks up the flow. We hope the changes we have made address the core of the reviewer’s concern.

The other striking difference between the artificial and natural images is the extreme regularity of the textures in the artificial images (at least of the plaids shown in Figure 2). The authors also used 1/f noise as a texture in the artificial images – did human performance differ depending on whether the artificial stimuli were plaid or 1/f noise? In general, it seems that adding more types of variation to the artificial stimuli and assessing the effects of that variation would provide a good way to assess what sorts of variation make human performance look more like it does for natural images. I suspect that the authors plan to do this in future work, but I think it would substantially increase the impact of this work to include such data and analyses here.

Human performance did not differ appreciably depending on the types of texture in artificial stimuli. Please see the plot above in Essential Revision #2. The variance of tilt estimates is slightly higher with 1/f stimuli, but the qualitative patterns are consistent across all three artificial stimulus textures.

We agree that studying the effects of the variation is an important question to address and we plan to do in the future. We believe a proper treatment of this question is a major undertaking in its own right and deserves its own manuscript.

Finally – I admit some disappointment with the choice of only showing 3 degree stimuli. To me, this lessens the impact of the work as well; a 3 degree image patch hardly constitutes a "scene". Thus, the following conclusion statement seems a bit of a reach: "We quantify performance in natural scenes and report that human tilt percepts are often neither accurate nor precise". Human estimates of tilt given full natural images (including much more context) would likely be better than the estimates reported here. I realize this is a very difficult problem, but eLife is also a broad, prestigous journal; studying tilt estimation in natural image patches may be a critical step on the way to studying tilt estimation in full scenes, but it also seems less broadly interesting.

Please see above Essential Revision #3. We have collected a new dataset with full scene stimuli, as the reviewer requested. We show the data from this new experiment in new figure (Figure 2) and discuss the result in a new Discussion section titled “Influence of full-field viewing”. Results are essentially unchanged with full-field viewing. While this result may seem surprising at first, it makes sense. Scene structure is correlated only over a relatively local area. Except for the ground plane, it is fairly unusual for surfaces to have a constant orientation over very large visual angles. Thus, scene locations farther than 3º are likely to add little additional information.

Last, a few notes on the model: First, I am puzzled as to why the authors do not include model performance on their artificial stimuli, too. This seems to be a straightforward and easy test of the generality of the model.

With artificial stimuli, the human estimate means are nicely predicted by the model, but the human estimates have higher circular variance than the model predicts, although the pattern is similar (see figure below). We do not yet understand the reason for this discrepancy, but we suspect that an explicit model of internal noise will be required. This is a topic we would like to reserve for future work.

**Author response image 1. respfig1:** Model performance on artificial stimuli

Second, it's not clear to me whether the "estimate cube" of optimal mappings between image cues and tilts is computed using some, all, or none of the same images that the subjects saw in the experiment. The authors should clarify this point.

The estimate cube includes the same scene locations that were presented in the experiment. However, the estimate cube was constructed from approximately 1 billion samples. Only 3600 of these samples were used as experimental stimuli, a negligible fraction of this total. Excluding the 3600 unique experimental stimuli from the estimate cube has no measurable influence on the model predictions. We now make this point in the Figure 5 caption.

I should note again that the concerns above are almost entirely about impact. I hesitate to reject an otherwise interesting and well-executed study on grounds that it's just not splashy enough. And there are several interesting and solid results in this paper. The fact that tilt variance is correlated with tilt angle in a large sample of natural scenes seems solidly supported and important. Modulo the questions I raised above, the MMSE model performance appears to provide a good match for human performance in the natural images. The persistent difference between errors estimating cardinal and oblique tilt, as well as the persistent bias to estimate horizontal tilt – both with matched tilt variance – are also interesting. Thus, I am on the fence about this paper, mostly because its impact seems marginal. I could be convinced to accept the paper with revisions or to reject the paper.Reviewer #2:This is a lovely paper, showing that a nonparametric Bayesian model of tilt estimation accounts startlingly well for human behavior in a tilt-estimation task. My comments are mainly about improving the clarity, not much more.Introduction: Many of my comments are a result of reading it in (my) natural order, i.e., your page order with diversions to the Methods when needed. So, when I got here I wondered whether the patches to be judged were centered on the display or occluded in the position in the original images. That's never stated explicitly but implied by a figure that hasn't come up yet.

Thanks. We now state that the patches were displayed in their original positions.

Introduction: You never motivate/justify pooling over tilt sign until much, much later, and so I was surprised you threw information away from the start. I wondered about it again for Figure 3 where, given that you provide disparity, the tilt sign ambiguity from pictorial cues should be alleviated.

As you point out, it is true that the disparity cue can alleviate tilt sign ambiguity. We are currently working on a new model that makes use of it. Also, even though the disparity cue is provided in the stimuli, there are some sign confusions in the human response (e.g., see a weak pattern of data points on the lower-right quadrant in the Figure 2 scatter plot). These sign confusions complicate some of the data analyses. Furthermore, all three humans are remarkably consistent in their unsigned tilt estimation performance. That said these are important issues that we will tackle in our next piece of work.

Figure 2 et seq.: Why didn't you run the model on the artificial stimuli and show the model fits for those data points (or misfits, as the case may be)?

Please see response above.

Subsection “Normative model”: The citation of Figure 6—figure supplement 2 here seems out of place. The analyses for this figure don't appear until the next page.

Thank you. We reorganized the text so that the material appears in a more natural order. The old Figure S6 is now Figure 6—figure supplement 2, and it is referred to only after introducing the analysis of trial-by-trial errors.

Also, shouldn't all the supplementary figures be cited somewhere in the main text? I think a bunch aren't.

All of the supplementary figures were cited in the main text.

Subsection “Trial-by-trial Error: Is -> Are.

Thank you. Fixed.

Figure 8:Exactly what bin cutoffs did you use for blue vs. red here?

Sorry. Tilt bins were 45º wide. For cardinal tilts, bins were centered on 0º and 90º. For oblique tilts, bins were centered on 45º and 135º. This information has now bin added to the Figure 8 caption.

Subsection “Effect of Natural Depth Variation”: artificially -> artificial.

Fixed.

Subsection “Generality of Conclusions and Future Directions”: our -> are.

Fixed.

Subsection “Cue-combination with and without independence assumptions”: This reference to Figure 6—figure supplement 2, since the pooled/averaged model is not in the figure, but merely mentioned in its legend.

Fixed. We removed the confusing reference.

Subsection “Experiment”: More details please: What's your definition of contrast.

Luminance contrast was defined as the root-mean-squared luminance values within a local area weighted by a cosine window. Specifically, luminance contrast is C=[∑x∈A((I(x)−I¯)/I¯)2W(x)]/∑x∈AW(x) where x is the spatial location, W is a cosine window with area A, and I¯=[∑x∈AI(x)W(x)]/∑x∈AW(x)is the local mean intensity.

Refer to the figure to state what part of the patch they were supposed to judge.

Observers estimated the tilt at the center of the 1º (or 3º) patch marked by the smaller of the two probe circles. This is indicated in the Figure 2 caption.

Were the judged bins over 180 or 360 degrees (I only say this because Figure 1 leads the reader to believe that it's over 180 degrees only).

Observers estimated tilt across all 360º. The groundtruth were sampled from 24 bins, each with a width of 15º. Each bin has 150 samples. We analyzed all the data but focused the majority of our analyses on the unsigned tilts (i.e., 360º modulo 180º).

Please show and give details about the two types of artificial stimuli. Do responses to them differ from one another?

We generated (1) 1/f textured plane, (2) a “sparse” 3.5cpd plaid plane (i.e., a plane textured with the sum of two orthogonal sinusoidal gratings), and (3) a “dense” 5.25cpd plaid plane. These details are included in the main text. We now show examples of all three artificial stimulus types in new Figure 2. In the same figure, we also show performance for each artificial stimulus type separately. Performance with all three stimulus types is similar. Note that all artificial stimuli were matched to the tilt, slant, distance, and luminance contrast of each patch of natural scene.

Subsection “Groundtruth tilt”: "atan2" is MATLAB notation, I'd think. You might want to say what you mean there.

Thanks. The correction has been made.

Subsection “Image cues to tilt”: I'd like more detail here as well. The disparity cue must be based on a definition of "local" and a restriction of cross-correlation shifts. The disparity gradient requires a scale.

Disparity was estimated using windowed cross correlation. The window for the windowed cross-correlation had the same space constant as the derivative operator used to compute the gradient. This information has been added to the methods section.

The disparity gradient doesn't have a tilt sign ambiguity, but the texture cue does.

Correct. The disparity gradient does not have a tilt sign ambiguity, but in this paper we focused only on recovering the unsigned tilt. We are currently working on generalizing the model so that it can predict signed tilt. In the future work, we will use the signed tilt information provided by the disparity gradient.

I'm not sure "patch size at half height" won't confuse people (you don't mean the viewed patch, but rather the patch after multiplying by the Gaussian window).

Fixed.

Figure 6—figure supplement 2: Actually, I'm rather surprised that the single-cue models (especially those other than disparity) perform as well as they do. It worries me that there are weird regularities in your database. You never say what your definition of luminance is exactly, but why should tilt be dependent on luminance?

Why does the orientation of the luminance gradient carry information about tilt? We don’t have a solid grasp of the physics, but it was been previously reported (Potetz & Lee, 2003) that luminance and depth is weakly correlated. But we don’t really know.

The luminance signals that we used for the computations were proportional to candelas/m^2^. Luminance contrast does not depend on the absolute luminance.

How are each cue binned?

Each cue is binned into 64 unsigned tilts, and the same bins are used for the single-cue models. Thus, three-cue model is binned 64^3^ bins.

Are the same bins used for the single-cue models, so that those models have vastly smaller measured parameters?

Yes, the single-cue models had fewer bins (i.e., fewer parameters). Increasing the number of single-cue bins does not improve performance. In other words, cue quantization error is not responsible for the single-cue performance.

Figure 6—figure supplement 2 legend, line 12: "… but better than the prior or…".

Fixed.